# Effects of Long-Term Physical Activity and BCAA Availability on the Subcellular Associations between Intramyocellular Lipids, Perilipins and PGC-1*α*

**DOI:** 10.3390/ijms24054282

**Published:** 2023-02-21

**Authors:** Vasco Fachada, Mika Silvennoinen, Ulla-Maria Sahinaho, Paavo Rahkila, Riikka Kivelä, Juha J. Hulmi, Urho Kujala, Heikki Kainulainen

**Affiliations:** Faculty of Sport and Health Sciences, NeuroMuscular Research Center, University of Jyväskylä, FI-40014 Jyväskylä, Finland

**Keywords:** lipid droplets, PLIN2, PLIN5, skeletal muscle, physical activity, C2C12, electrical pulse stimulation, EPS, subcellular localization

## Abstract

Cellular skeletal muscle lipid metabolism is of paramount importance for metabolic health, specifically through its connection to branched-chain amino acids (BCAA) metabolism and through its modulation by exercise. In this study, we aimed at better understanding intramyocellular lipids (IMCL) and their related key proteins in response to physical activity and BCAA deprivation. By means of confocal microscopy, we examined IMCL and the lipid droplet coating proteins PLIN2 and PLIN5 in human twin pairs discordant for physical activity. Additionally, in order to study IMCLs, PLINs and their association to peroxisome proliferator-activated receptor gamma coactivator 1-alpha (PGC-1α) in cytosolic and nuclear pools, we mimicked exercise-induced contractions in C2C12 myotubes by electrical pulse stimulation (EPS), with or without BCAA deprivation. The life-long physically active twins displayed an increased IMCL signal in type I fibers when compared to their inactive twin pair. Moreover, the inactive twins showed a decreased association between PLIN2 and IMCL. Similarly, in the C2C12 cell line, PLIN2 dissociated from IMCL when myotubes were deprived of BCAA, especially when contracting. In addition, in myotubes, EPS led to an increase in nuclear PLIN5 signal and its associations with IMCL and PGC-1α. This study demonstrates how physical activity and BCAA availability affects IMCL and their associated proteins, providing further and novel evidence for the link between the BCAA, energy and lipid metabolisms.

## 1. Introduction

On top of being the largest organ in the human body, skeletal muscle has high energy demands, leading to elevated lipid turnover rates [1,2,3]. Despite being well established, the connection between skeletal muscle lipid metabolism and metabolic health is far from linear. On one hand, several metabolic diseases—such as insulin resistance—have been associated with physical inactivity and elevated intramyocellular lipids (IMCL). On the other hand, highly insulin sensitive individuals—such as endurance athletes—associate with even higher levels of IMCL [2,4]. It became ever more clear that mere levels of IMCL were not sufficient to explain skeletal muscle lipid metabolism efficiency. The perilipin protein family members (PLINs) are central agents in managing the fate of IMCL, and they are mostly known for regulating the access of lipolytic and lipogenic enzymes onto the surface of lipid droplets (LDs), thus protecting the cell against lipotoxicity and improving metabolic health [5,6,7,8,9,10].

The responses of human skeletal muscle LDs and PLINs to different exercise modalities have been well studied in a multitude of setups [11,12,13,14,15]. However, up to this date, the effects of long-term physical activity on IMCL and PLINs, amongst genetically similar individuals, remain largely unstudied. As a first aim of the present work, we propose to explore this gap. Therefore, through confocal microscopy and pixel-to-pixel intensity correlation analysis (ICA) [16], we examined IMCL, PLIN2 and PLIN5 and their associations in two main fiber types of twin pairs with discordant physical activity. We hypothesize that physically discordant twins will display different patterns of IMCL-PLIN association relative to previously studied setups.

The IMCL-PLINs dynamics are complex, and the role of PLINs themselves is not limited to hydrolysis or esterification of triacylglycerol (TAG) within LDs. For instance, PLIN5 has been shown to be necessary to transport monounsaturated fatty acids (MUFAs) into nuclei, in order to activate peroxisome proliferator-activated receptor gamma coactivator 1-alpha (PGC-1α) and, consequently, induce mitochondria biogenesis and fatty acid oxidation [17,18]. Additionally, PGC-1α is also an important bridge between the skeletal muscle lipid and branched-chain amino acid (BCAA) metabolisms, as it activates the latter through multiple nuclear receptors [19,20]. Importantly, unlike other amino acids that can be processed by the liver, BCAA are mostly catabolized in skeletal muscle [21,22]. Respectively, we have previously hypothesized that the connection between IMCL and metabolic health could further extend to the BCAA metabolism [22], as one important source of TAG in muscle comes from glyceroneogenesis [23]. Furthermore, inefficient skeletal muscle BCAA metabolism has been associated with impaired lipid metabolism and insulin resistance [21]. Finally, there is a known interplay between exercise and muscle BCAA metabolism [24,25]. However, studies establishing the relationship between intramyocellular BCAA and PLINs are essentially lacking. Therefore, it is important to investigate the impacts that BCAA availability may have on LD-PLINs regulation.

As a second aim, we investigated IMCL, PLIN2, PLIN5 and PGC-1α subcellular responses to exercise and BCAA availability. By combining electrical pulse stimulation (EPS)—an exercise-mimicking method [26]—with BCAA deprivation, we measured the optical density and performed ICA in different myotube compartments. We hypothesize that skeletal muscle PLIN5 and PGC-1α signals associate in response to EPS, especially within the nuclei. Moreover, we postulate that, besides LDs, PLIN5 or PGC-1α, PLIN2 may have unreported nuclear associations, which could be affected by EPS and/or BCAA deprivation.

## 2. Results

### 2.1. Active Co-Twins Have Increased IMCL in Type I Fibers

Concerning the twin participants described in Table 1, type I fibers contained more IMCL than type II fibers, as expected (*p*
<0.001, Figure 1A–C). Interestingly, physically active twins had increased IMCL in type I fibers compared to their inactive co-twin (*p* < 0.001, Figure 1B,C), but there was no difference in type II fibers due to LTPA. The active twins demonstrated a significant difference in IMCL between fiber types (*p*
<0.001), which was not observed in the inactive co-twins (*p*
=0.064), as seen in Figure 1C.

As expected, PLIN5 associated IMCL was significantly higher in type I than in type II fibers (*p*
=0.001, Figure 2A,B), and with no differences between twin pairs (Figure 2B,C). Lastly, both *PLIN5* mRNA levels and PLIN5 confocal mean signal remained unchanged between twin pairs (Appendix A).

### 2.2. Inactive Twins Show Decreased IMCL-PLIN2 Association

The inactive twin pairs show a decreased association between IMCL and PLIN2 (*p*
=0.008), mainly through a very significant decrease in the type II fibers (*p*
<0.001, Figure 3A–C). The latter happened despite no fiber type or LTPA differences in PLIN2 mean signal (Appendix A) or *PLIN2* mRNA levels (Appendix A). Taken together, this shows that IMCL targeting by PLIN2 is clearly restrained in type II fibers of inactive twins.

### 2.3. PLIN5 Abounds in C2C12 Myotube Nuclei, PLIN2 Detected

Next, given the role in regulating energy metabolism and a known nuclear interaction with PLIN5, we studied PGC-1α and its association with IMCL and PLIN5 in different myotube compartments.

The compartmental analysis showed a significant contrast (*p*
<0.001) between cytosolic and nuclear signals for all markers within the myotubes. The most abundant nuclear signals were observed for IMCL and PLIN5. A much smaller proportion of PLIN2 (*p*
<0.001) was detected above the background and occasionally in a particle-like manner (Figure 4A,B).

### 2.4. PLIN2 Dissociates from IMCL upon BCAA Deprivation in Myotubes

Both IMCL and PLIN2 showed a mostly diffused signal in C2C12 myotubes. Occasionally, semi-spherical IMCL aggregates were visible as LDs (Appendix A). Likewise, PLIN2 aggregates were common and often seen as dotted ring structures enveloping LDs (Figure 5A).

In addition to fiber type and exercise, BCAA can also affect IMCL metabolism, and this may interact with muscle contraction. Respectively, we observed a cytosolic decrease in the association between PLIN2 and IMCL after BCAA deprivation (*p* =0.028, Appendix A), especially after EPS (*p*
=0.048, Figure 5B). The same combination (No BCAA|EPS) resulted in increased PLIN2 and PLIN5 association inside the nuclei (*p*
=0.030), and in a dependent manner (*p*
=0.033, Figure 5C).

Such events were independent from the overall PLIN2 signal, which remained unchanged after EPS and BCAA deprivation (Appendix A).

### 2.5. PLIN5 Moves to Myotube Nuclei upon Stimulation, Further Associating with IMCL and PGC-1α

The PLIN5 signal was mostly punctate and abundant, often immediately adjacent to or colocalizing with other markers. In turn, PGC-1α showed mostly a diffused signal, sometimes concentrating in differently shaped aggregates and often colocalizing with IMCL (Figure 6A).

Notably, PLIN5 signal increased in nuclei after EPS (*p* =0.033, Figure 6B), where it associated with IMCL (*p*
=0.019, Figure 6D), especially under BCAA availability (*p*
=0.002, Appendix A). Likewise, under Normal BCAA|EPS, nuclear PLIN5 further associated with PGC-1α, and very significantly so (*p*
=0.009, Figure 6E).

Interestingly, BCAA deprivation alone was sufficient to decrease PGC-1α signal in myotube nuclei (*p*
=0.009, Figure 6C). It is worth noting that such effect was seen only in the nuclei, as neither a cytosolic confocal signal (Figure 6B) nor whole cell Western blots detected changes in PGC-1α protein concentration (Appendix A).

In the cytosol, the level of association between PLIN5 and PGC-1α was increased after BCAA deprivation (*p*
=0.026), as seen in (Figure 6E). The level of association between IMCL and PGC-1α was not altered with either EPS or BCAA deprivation (Appendix A).

## 3. Discussion

### 3.1. Overview

This study examined the effects of physical activity on intramyocellular lipids and respective coating proteins in human twin pairs discordant for life-long physical activity. We found that, in physically active twins, the intramyocellular phenotype resembles that of athletes, namely in their type I fiber elevated lipid content, together with an enhanced lipid coating by PLIN2.

Secondly, we investigated myotube inter-compartmental responses to muscle contraction induced by EPS and to BCAA deprivation. We found that BCAA deprivation leads to a cytosolic dissociation between PLIN2 and IMCL, especially when combined with EPS. Importantly, we found that EPS leads to an increased presence of PLIN5 in nuclei, with increased association to PGC-1α, IMCL and PLIN2. Finally, we found that the signal of nuclear PGC-1α is abruptly decreased after BCAA deprivation.

### 3.2. Active Twins Resemble Athlete Phenotype

It has been shown that a healthy elevation of IMCL can be expected from not only athletes [2,4,12], but also from sedentary individuals who underwent a 6-week training period, especially in type I fibers [13]. Although the overall IMCL content was not different between twin pairs, we did observe significantly increased IMCL in type I fibers of active twins when compared to their inactive co-twins (Section 2.1). Our results suggest that the *athlete paradox* phenotype [2] may not be genetically determined and might be reached via life-long LTPA. Concomitantly, we have previously demonstrated that the active twins have improved skeletal muscle oxidative energy and lipid metabolism [24]. It should be noted that LTPA has recently been associated with slower epigenetic aging [27], suggesting that such mechanisms may be behind the results reported in our work.

Of the muscle PLINs, PLIN5 is probably the most studied member, and it is known for positively responding to exercise and high fat diet, both at the protein level and on IMCL association [8,9,12,28,29,30]. Interestingly, despite an obvious fiber type difference, LTPA led to no changes in PLIN5 signal or PLIN5 associated IMCL (Section 2.3). This may reflect the fact that intramyocellular physiological responses driven by LTPA could be distinct from those of more strenuous exercise programs in previous studies.

In addition, associating with efficient TAG storage and healthier profiles, intramyocellular PLIN2 has been shown to increase with exercise [13,31]. Although we did not register changes in PLIN2 signal, we did observe a significant decrease in IMCL-PLIN2 association in type II fibers of inactive twins (Section 2.2), suggesting an unhealthier phenotype. The hypothesis that lipotoxic signaling in skeletal muscle could originate from type II fibers is not new, as this fiber type is generally ill-equipped to metabolize lipids [32], especially in innermost regions of fibers ([33], Appendix A). Future studies should further explore the signaling impact of poorly PLIN coated-IMCL in glycolytic muscle fibers.

### 3.3. Myotubes Resembling Type II Fibers

Beyond PLINs [31], exercise and EPS are also expected to increase PGC-1α levels in skeletal muscle, including C2C12 myotubes [34,35,36,37]. Associated with mitochondrial biogenesis and fatty acid oxidation, as well as with glucose uptake and decreased glucose oxidation, PGC-1α is a rather lipolytic and glucogenic agent [38]. However, in the current study, EPS alone did not trigger significant cytosolic responses in the signal of PLIN2, PLIN5, PGC-1α or their association. Accordingly, from previous studies using the same protocol, we have observed a sharp glycolytic response in C2C12 myotubes [25] and only a modest lipolytic one [39]. More specifically, we had shown that EPS led to unchanged IMCL content, unaffected lipogenesis and decreased lipid oxidation [39].

One study has reported increased PGC-1α protein and unchanged lactate or pyruvate levels after using EPS [40]. Contrastingly, we have observed unchanged PGC-1α signal (this study) and increased lactate and pyruvate-derived products [25]. Interestingly, pyruvate is a known inhibitor of PGC-1α [41] and could be hindering a stronger lipolytic response in our setup. Conflicting results when studying lipid metabolism in C2C12 are not uncommon, as the generally glycolytic nature of this cell line can be increased with longer differentiation protocols [40,42]. Future research should focus on the same phenomena using different cell lines and culture parameters.

### 3.4. BCAA Necessary for PLIN2 Coating of IMCL

Our group has earlier demonstrated that the unhealthier profile of the inactive twins extends beyond an inefficient lipid metabolism, showing an associated downregulation of BCAA catabolism [24]. Furthermore, we have previously shown that BCAA deprivation decreases both lipid oxidation and lipogenesis in myotubes. In addition, when combined with EPS, BCAA deprivation also decreased the number of segmented LDs [39]. In the current work, the latter combination resulted in a dissociation between cytosolic IMCL and PLIN2 (Section 2.4), while the association between IMCL and PLIN5 remained unchanged (Section 2.5).

Often associating with TAG accumulation and protection against lipotoxicity derived-insulin resistance, PLIN2 is known to abound on the surface of LDs. There it can bind to both lipases and esterification enzymes, possibly having a more lipogenic role than PLIN5 [5,6,31,43]. Finally, there is evidence suggesting that BCAA facilitates TAG accumulation in muscle [44] and that EPS increases BCAA catabolism [25]. Together with our results, these data suggest that PLIN2 coating of IMCL is central to a healthy storage of lipid derivatives, which is improved by BCAA availability combined with physical activity, ultimately resulting in efficient BCAA and lipid metabolisms.

An efficient BCAA catabolism together with BCAA availability promotes ketonegenesis via several mitochondrial enzymes [45], some of which can further synthesize cholesterol for cellular needs [46]. It can be speculated that, in a scenario of impaired BCAA catabolism or BCAA deprivation, the source of ketone bodies and cholesterol could shift towards IMCL, once exposed and uncoated from PLIN2 (Figure 7A,C). It is known that C2C12 myotubes are able to produce ketone bodies, especially when contracting [25] and that cholesterol is a major constituent of IMCL [47].

### 3.5. Setting Transcription in Cytosol?

In this study, the individual signals from cytosolic PLIN5 and PGC-1α remained unaltered after BCAA deprivation. Nevertheless, the same experiment led to an increase in the cytosolic association between these markers (Section 2.5), often with very clearly colocalizing aggregates (Figure 7B). The same two proteins are known to interact, even if not directly binding [17]. Moreover, PGC-1α can interact with nuclear receptors, such as estrogen-related receptor α (ERRα) and sirtuin 1 (SIRT1), which have also been shown to be present in cytosolic pools [48,49] (Figure 7C). Interestingly, PGC-1α-mediated upregulation of BCAA metabolism does require ERRα [20], whose activation is dependent on cholesterol [50].

The relationship between cytosolic ERRα-PGC-1α-SIRT1-PLIN5-IMCL complexes and BCAA availability is unclear. It is possible that, through PLIN5 coordination, cytosolic IMCL could provide cholesterol and MUFAs to ERRα and SIRT1, respectively (Figure 7C), thus triggering the translocation to nuclei and consequent activation of transcription factors. In fact, we observed that, under Normal BCAA, EPS led to an increased triple association between cytosolic PGC-1α-PLIN5-IMCL (Appendix A). More research is needed to clarify the dynamics between cytosolic pools of IMCL, PLINs, and related transcription factor co-activators.

### 3.6. Nuclear Affairs

Gallardo-Montejano et al. have demonstrated that catecholamine and fasting causes phosphorylation of PLIN5 and its translocation to nuclei, promoting PGC-1α activation via SIRT1 disinhibition [17]. The same authors hypothesized that exercise would cause a similar response. Our results support this hypothesis, as we unprecedentedly showed that, in response to EPS, PLIN5 does enrich myotube nuclei, further associating with PGC-1α (Section 2.5). More recently, Natj et al. elegantly demonstrated that PLIN5’s role in activating SIRT1-PGC-1α occurs via MUFAs binding and chaperoning from LDs towards the nuclei [18]. Although we observed an increase in both IMCL-PLIN5 and PGC-1α-PLIN5 associations in the nuclei, we did not observe any changes in IMCL-PGC-1α nuclear association (Appendix A). This suggests that the delivery of MUFAs by PLIN5 does not require close proximity between nuclear IMCL and PGC-1α aggregates, and that nuclear IMCL-PLIN5 association possibly has further roles to be explored.

Independent of EPS, we have observed a very sharp decline in nuclear PGC-1α after BCAA deprivation (Section 2.5). This may indicate that the function of PGC-1α as a nuclear transcription factor co-activator becomes hampered by BCAA deprivation. This is consistent with the decreases in lipid oxidation and lipogenesis we have previously observed from the same setup and experiment [39]. The precise mechanism behind such strong effects cannot be demonstrated by our study, but BCAAs are known inducers of the mammalian target of rapamycin complex 1 (mTORC1) [51], which in turn is an important activator of PGC-1α [52] (Figure 7B).

Lastly, LDs and other PLINs have also been reported from nuclei. While nuclear PLIN3 and PLIN5 were shown abundant, nuclear PLIN2 has been reported as virtually absent and largely unresponsive [17,53]. Our results corroborate the latter observation (Section 2.3). However, we observed that, when electrically stimulated, PLIN2 increasingly associates with PLIN5 in myotube nuclei, especially if deprived from BCAA (Section 2.4). Although it is believed that PLIN2 and PLIN5 do not bind each other [17,54], their spatial correlation under such stimuli may indicate some level of indirect interaction, leading to possible changes in the metabolic status. Future research is needed on potential coordinating roles between the different PLINs within nuclei.

### 3.7. Limitations and Strengths

Although a strong model to study effects of physical activity even in small cohorts [24,55], the number of twins and respective fibers were relatively small, especially for studying such dynamic events which tend to be very variable between cells. Despite this limitation, concerning IMCL and PLINs, this is the first study showing the effects of life-long physical activity in genetically similar humans, bringing additional insights to the area.

While it is our strength that we have two very different models of physical activity/exercise, the C2C12 and twin models used in this study should not be directly compared. In addition, the current C2C12 experimental model may be too glycolytic for robust IMCL metabolism studies. However, knowing that undifferentiated myoblasts are too far removed from muscle biology, we were able to classify and measure the signal from myotubes exclusively (Appendix A), producing novel observations. Nevertheless, future studies should try to replicate the same observations in other cell models, such as FACS sorted myofibers or satellite cells.

One of the focuses of this work was to investigate if BCAA deprivation would affect IMCL, PLINs and PGC-1α, thus establishing this unexplored link. We have not, however, studied the effects of BCAA over-supplementation on these markers and, therefore, this should be addressed in future research.

Moreover, given the suboptical nature of diffused biomolecules, we preferred to focus on overall signal intensity rather than solely on thresholded objects—hence, for instance, the focus on the term IMCL, more inclusive and not limited to LDs or TAG. With the current instruments, we could have confidently segmented the brighter marker aggregates of about ø 1 μm, but by doing so, we would be discarding precious information coming from smaller or more diffused aggregates (Figure 7A).

In our study, we cannot strictly pertain to protein–protein or IMCL–protein binding interactions, something that could be aided by fluorescence resonance energy transfer, cell fractionation or immunoprecipitation. Instead, we conducted ICA on distinct tissue and intracellular compartments, allowing us to investigate the distribution and spatial correlation between multiple markers, some of which are not immunopercepitable. Even when two given biomolecules do not directly bind, the statistical correlation between their signals can inform us about regions where such markers could—either directly or indirectly—associate and co-act on given signaling phenomena [16,56].

## 4. Materials and Methods

### 4.1. Human Twin Pairs

A total of 8 participants from 4 same-sex twins pairs (2 male and 2 female) with discordant leisure time physical activity (LTPA) for 32 years were identified from the Finnish Twin Cohort (Table 1). Discordance was based on a series of structured questions concerning leisure activity and physical activity during journeys to and from work. The leisure time (metabolic equivalent (MET)) index was calculated by assigning a multiple of the resting metabolic rate to each form of physical activity (intensity × duration × frequency) and expressed as a sum score of leisure time MET hours per day. It is worth noting that the active twins’ average LTPA score (13.8, Table 1), roughly corresponds to 1 h of running per day, for more than three decades. On the other hand, the inactive twins are not sedentary and endured basic levels of LTPA.

The study participants were advised not to exercise vigorously during the morning and two days before both of their laboratory visits (one visit for clinical examinations including exercise tests and one visit for biopsy studies). Muscle tissue samples were taken after an overnight fast between 8:00 a.m. and 10:00 a.m. under local anesthesia after skin cooling and disinfection. Using a suction technique with a Bergström’s needle (ø 5 mm), the muscle biopsy was taken from *Vastus lateralis* at the midpoint between *Trochanter major* and the lateral joint line of the knee. The sample was then mounted transversely on cork with Tissue Tek™ (Miles, Elkhart, In, USA; Sakura, Cat. # 4583), and frozen rapidly (10–15 s) in isopentane (Fluka, Cat. # 59080 ), precooled to −160 °C in liquid nitrogen and stored at −80 °C.

For further details on participant description and recruiting procedures, see Leskinen et al. (2009, 2010) [24,55].

### 4.2. Myotube Experiments

Murine C2C12 myoblasts (American Type Culture Collection, ATCC, Manassas, VA, USA) were maintained in high glucose-containing Dulbecco’s Modified Eagle growth medium (GM) ($4.5\;g\;\cdot \;L^{-1}$, DMEM, #BE12-614F, Lonza, Basel, Switzerland) supplemented with 10% (*v*/*v*) fetal bovine serum (FBS, #10270, Gibco, Rockville, MD, USA), $100\;U\;\cdot \;{\mathrm{mL}}^{-1}$ penicillin, $100\;\mu g\;\cdot \;{\mathrm{mL}}^{-1}$ streptomycin (P/S, #15140, Gibco) and 2 mM L-Glutamine (#17-605E, Lonza, Basel, Switzerland). Myoblasts were seeded on 6-well plates (NunclonTM Delta; Thermo Fisher Scientific, Waltham, MA, USA). When the myoblasts reached 95–100% confluence, the cells were rinsed with phosphate-buffered saline (PBS, pH 7.4), and the GM was replaced by differentiation medium (DM) containing high glucose DMEM, 2% (*v*/*v*) horse serum (HS, 12449C, Sigma-Aldrich, St. Louis, MO, USA), $100\;U\;\cdot \;{\mathrm{mL}}^{-1}$ and $100\;\mu g\;\cdot \;{\mathrm{mL}}^{-1}$ P/S and 2 mM L-glutamine to promote differentiation into myotubes. Fresh DM was changed every second day. The cells were screened negative for mycoplasma contaminations, following manufacturer’s instructions (MycoSPY Master Mix Test Kit, M020, Biontex, München, Germany). The experiments were conducted on days 5–6 post differentiation and on a duplicated way.

The myotubes on 6-well plates were acclimatized to 0.1 mM oleic acid and 1 mM L-carnitine in normal BCAA DM on the day 4 post differentiation. On the next day, the electrodes were placed directly onto the wells. The electrical stimulation (1 Hz, 2 ms, 12 V) was applied to the cells using a C-Pace pulse generator (C-Pace EM, IonOptix, Milton, MA, USA) for 24 h at 37 °C with the same protocol as described earlier [25]. As described previously, EPS was paused after 22 h and target BCAA concentrations (no BCAA or normal BCAA) were employed to investigate the interactive effects of EPS and BCAA deprivation.

The BCAA deprivation experiments were carried out for 2 h at 37 °C in high-glucose BCAA-free DM (4.5 g · L^−1^, BioConcept, 1-26S289-I, Allschwill, Switzerland). The experimental groups were as follows: (1) cells supplemented with 0.8 mmol · L^−1^ of all BCAAs without EPS (Normal BCAA|Rest) or (2) with EPS (Normal BCAA|EPS), and (3) cells deprived (0.0 mmol · L^−1^) of all BCAA’s without EPS (No BCAA|Rest) or (4) with EPS (No BCAA|EPS).

### 4.3. Protein Extraction and Western Blotting

The C2C12 cells were harvested and Western blotting was conducted as previously described [25] with minor modifications. Briefly, 10 μg of total protein per samples were loaded on 4–20% Criterion TGX Stain-Free protein gels (#5678094, Bio-Rad Laboratories, Hercules, CA, USA) and samples were separated by SDS-PAGE.

To visualize proteins using stain-free technology, the gels were activated and the proteins were transferred to the PVDF membranes. Membranes were blocked with Intercept Blocking Buffer (#927-70001, LI-COR, Lincoln, NE, USA) followed by overnight incubation at 4 °C with primary antibody (PGC-1α, 1:10,000, ab191838, Abcam, Cambridge, UK) in Intercept Blocking Buffer diluted (*v*:*v*, 1:1) with Tris-buffered saline (TBS) with 0.1% Tween-20.

Membranes were incubated with the horseradish peroxidase-conjugated secondary IgG antibody (anti-Rabbit, 1:4,0000) (Jackson ImmunoResearch Laboratories, West Grove, PA, USA) in Intercept Blocking Buffer diluted (*v*:*v*, 1:1) with TBS-0.1% Tween 20. Enhanced chemiluminescence (SuperSignal west femto maximum sensitivity substrate; Pierce Biotechnology, Rockford, IL, USA) and ChemiDoc MP device (Bio-Rad Laboratories) were together used for protein visualization.

Stain free (75-250 kDA area of the lanes) was used as a loading control and for the normalization of the results.

### 4.4. Gene-Expression Arrays

The RNA preparation, cRNA generation and microarray hybridization procedures were used as previously described [24]. In brief, Trizol-reagent (Invitrogen, Carlsbad, CA, USA) was used to isolate total RNA from the twin muscle biopsies, which were homogenized on FastPrep FP120 apparatus (MP Biomedicals, Illkirch, France).

An Illumina RNA amplification kit (Ambion, Austin, TX, USA) was used according to the manufacturer’s instructions to obtain biotinlabeled cRNA from 500 ng of total RNA.

Hybridizations to Illumina HumanWG-6 v3.0 Expression BeadChips (Illumina Inc., San Diego, CA, USA) containing probes for *PLIN2* and *PLIN5*, were performed by the Finnish DNA Microarray Center at Turku Center for Biotechnology according to the Illumina BeadStation 500x manual.

Hybridized probes were detected with Cyanin-3-streptavidin ($1\;\mu g\;\cdot \;{\mathrm{mL}}^{-1}$, Amersham Biosciences, GE Healthcare, Uppsala, Sweden) using Illumina BeadArray Reader (Illumina Inc.) and BeadStudio v3 software (Illumina Inc.).

The gene expression data and the raw data sets for skeletal muscle have been deposited in the *GEO* database (https://www.ncbi.nlm.nih.gov/geo/query/acc.cgi?acc=GSE20319, accessed on 5 January 2023).

### 4.5. Histology

For each twin, two 8 μm cross sections were made in a cryostat at −25 °C (Leica CM 3000, Wetzlar, Germany) and collected onto 13 mm round coverslips. For the cell culture experiments, duplicate 6-well plates were used containing three 13 mm round coverslips per well. Each experimental group was measured from 18 coverslips. After the 24 h of EPS, the plates were removed from the incubator and the medium aspirated. In both the human and C2C12 experiments, the samples were immediately fixed in 4% paraformaldehyde for 15 min at room temperature (RT). After washing for 3 × 5 min with PBS, the samples were blocked with 10% goat serum (GS) in PBS-0.05% saponin (PBSap) for 30 min and then washed briefly with PBSap. Primary antibodies were diluted in 1% GS-PBSap and incubated for 1 h at RT. A 3 × 10 min wash in PBSap ensued before incubating the secondary antibodies for 1 h at RT. Excess antibody was removed with another 3 × 10 min wash in PBSap. Finally, non-immuno stains were incubated for 30 min before 2 × 10 s washed with PBS. Thorough vial mixing and smooth rocking were ensured for every incubation step in order to grant an even stain.

For the twin studies, two different quadruple staining procedures took place, each sharing as common markers: LD540 for IMCL (0.1 μg· mL^−1^,[57]]), caveolin 3 for sarcolemma (2 μg· mL^−1^, PA1-066, Thermo Fisher Scientific) and slow myosin heavy chain (MyHC) for type I fibers (2 μg· mL^−1^, A4.951, DSHB, University of Iowa, IA, USA). While one section was further incubated for PLIN2 (1:200 dilution, GP47, Progen), the second section was instead incubated for PLIN5 (1:200 dilution, GP31, Progen). Respectively, the following secondary antibodies were used in combination: Alexa Fluor 405 Goat anti-Rabbit IgG (H+L), Alexa Fluor 594 Goat anti-Mouse IgG (H+L) and Alexa Fluor 488 Goat anti-Guinea Pig IgG (H+L) (Thermo Fisher Scientific).

For the C2C12 samples, a quintuple staining was performed, using 3 antibody markers: differentiated myotubes (5 μg· mL^−1^, MF-20, DSHB), PLIN5 (1:200 dilution, GP31, Progen), plus either PLIN2 (5 μg· mL^−1^, ab52356, Abcam) or PGC-1α (5 μg· mL^−1^, ab191838, Abcam). Respectively, the following secondary antibodies were used in combination: Alexa Fluor 647 Donkey anti-Mouse IgG (H+L), Alexa Fluor 488 Goat anti-Guinea Pig IgG (H+L) and Alexa Fluor 594 Donkey anti-Rabbit IgG (H+L) (Thermo Fisher Scientific). Additionally, IMCL (0.1 μg· mL^−1^, LD540, [57]) and nuclei (5 μg· mL^−1^, DAPI, Thermo Fisher Scientific) were stained in all coverslips. Cross reactivity was successfully ruled out by carefully controlling every antibody combination.

Every coverslip was mounted on microscopy slides using Mowiol with 2.5% DABCO (Sigma-Aldrich) and left to dry for 24 h in the dark at 4 °C. Imaging took place within 48 h after mounting.

### 4.6. Image Acquisition

All image data were acquired on a LSM700 confocal microscope using the *ZEN black* software (Zeiss, Germany). Twin data were collected with a Plan-Apochromat 20x/0.8 objective (Zeiss, Germany), producing 2 images per participant, each image covering an area of 320.1 × 320.1 μm (voxel size = 0.31 × 0.31 × 2.4 μm). Cell data were collected with a Plan-Apochromat 63×/1.4 oil objective (Zeiss, Germany) from 3 random 203.2 × 203.2 μm confluent areas in each coverslip, producing 9 images per well and a total of 54 images per experimental group (voxel size = 0.1 × 0.1 × 1 μm).

Multi channel acquisition was achieved through the use of 4 laser lines (405, 488, 555 and 639 nm). Bleed-through was successfully avoided in each channel by manually configuring the secondary dichroic mirror position over two different photomultiplier tubes. Control samples incubated solely with secondary antibodies were used to set background values.

### 4.7. Image Analyses

For C2C12, the MF-20 signal was used to segment and analyze differentiated myotubes only. The nuclei detected within the segmented myotubes were also segmented, thus allowing compartmental analyses on the cytosolic versus nucleic markers. For the human cross sections, only intact and artifact-free fibers were segmented (Active = 19.3 ± 2.5 and Inactive = 15.5 ± 4.3. Mean ± SE). Segmentation was aided by machine learning algorithms using the *Trainable Weka Segmentation* tool [58] in *Fiji* [59]. Each analyzed fiber cross section was classified into either type I or type II, according to the detected and thresholded signal of slow myosin per cell area. The optical density of each marker was determined by measuring the mean value of the respective signal within each cell. The level of association between the different markers was determined through pixel-to-pixel intensity correlation analysis (ICA), by thresholding image data according to Costes et al. [16] in *Fiji*. In order to ensure a zero valued background, prior to analyses, all images were denoised and deconvoluted in *Fiji* using a theoretical point spread function separately for each channel.

### 4.8. Data Cleaning and Statistics

For the twin data, given the low number of participants, statistical cases are constituted of individual muscle fibers. Concerning the C2C12 data, from each coverslip, the 2 closest values per variable were averaged, while from each well, the values from the 2 closest coverslips were further averaged. To control for inter cell batch variability, all values were normalized against the control group (Normal BCAA|Rest). Finally, for both human and C2C12 studies, outliers were identified and removed via z-score (2 standard deviations).

Boxes in the boxplot figures depict interquartile ranges and medians, while whiskers represent the 95% confidence interval, unless stated otherwise. The main effect significance is marked with **#** and combined group significance is marked with *****, while interacting effects between independent variables are expressed with **&**. Normality was assessed with Shapiro–Wilk tests and group comparisons were preformed with either Mann–Whitney U tests or *t*-tests, depending on data distribution. Interacting effects were tested with a two-way ANOVA. Given the large number of human muscle fibers, significance levels were set at *p* <0.01 and *p*
<0.001. For C2C12, the significance levels were set at *p*
<0.05 and *p*
<0.01. The ICA between the different markers were tested with a Manders split coefficent test after the thresholding step mentioned in Section 4.7.

Data crunching, statistics and boxplot visualization were performed in *Python 3.9.0*, with the packages *NumPy* [60], *pandas* [61], *SciPy* [62], *statsmodels* [63], *seaborn* [64] and *matplotlib* [65], respectively. All *Fiji* and *Python* routines can be found at https://github.com/seiryoku-zenyo/twinC2C12-studies, accessed on 1 August 2022.

Comparison and analysis of mRNA data were performed with the *GEO2R* functionality on the *GEO* database (https://www.ncbi.nlm.nih.gov/geo/geo2r/?acc=GSE20319, accessed on 5 January 2023).

Protein network analysis and visualization were performed with *STRING* [66] and *Cytoscape* [67] software.

## 5. Conclusions

The present study expands our basic knowledge on the known link between a functional BCAA metabolism, physical activity and an efficient lipid metabolism: specifically, via PLIN2’s function in coating IMCL upon BCAA availability and long-term physical activity. Moreover, we showed that PLIN5 has an ability to translocate to nuclei and associate with PGC-1α after contractions.

## Figures and Tables

**Figure 1 ijms-24-04282-f001:**
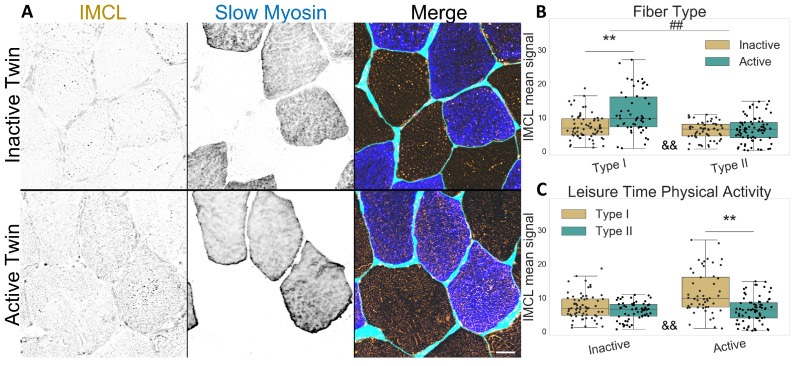
IMCL mean signal intensity between twin pairs. (**A**) representative image showing differences between groups. Gray level indicates signal, cyan indicates segmented sarcolemma. Note the active twin type I fibers with higher IMCL signal; Bar = 20 μm; (**B**) fiber type as main effect, with LTPA combined; (**C**) LTPA main effect, with fiber type combined; main effect differences denoted with **##** (*p*
<0.001); combined group differences denoted with ****** (*p*
<0.001); interacting effect between fiber type and LTPA denoted with **&&** (*p*
<0.001). Dots in (**B**,**C**) represent individual cells.

**Figure 2 ijms-24-04282-f002:**
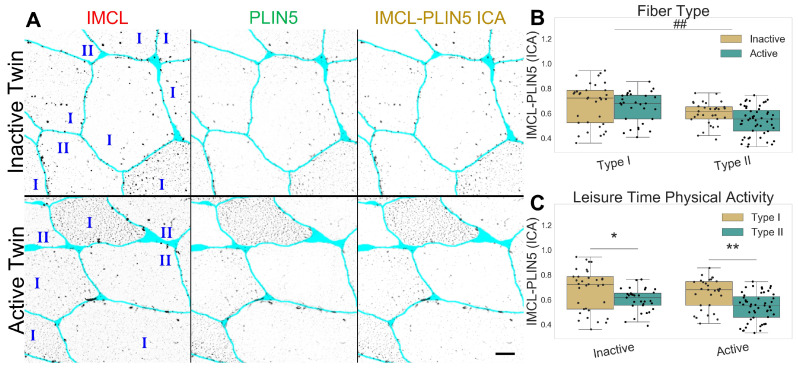
IMCL-PLIN5 intensity correlation analysis in twin pairs. (**A**) representative image showing differences between groups. Gray level indicates signal, cyan indicates segmented sarcolemma. Bar = 20 μm; (**B**) fiber type as main effect, with LTPA combined; (**C**) LTPA main effect, with fiber type combined. Main effect differences denoted with **##** (*p*
<0.001). Combined group differences denoted with ***** (*p*
<0.01) and ****** (*p*
<0.001). Dots in (**B**,**C**) represents individual cells.

**Figure 3 ijms-24-04282-f003:**
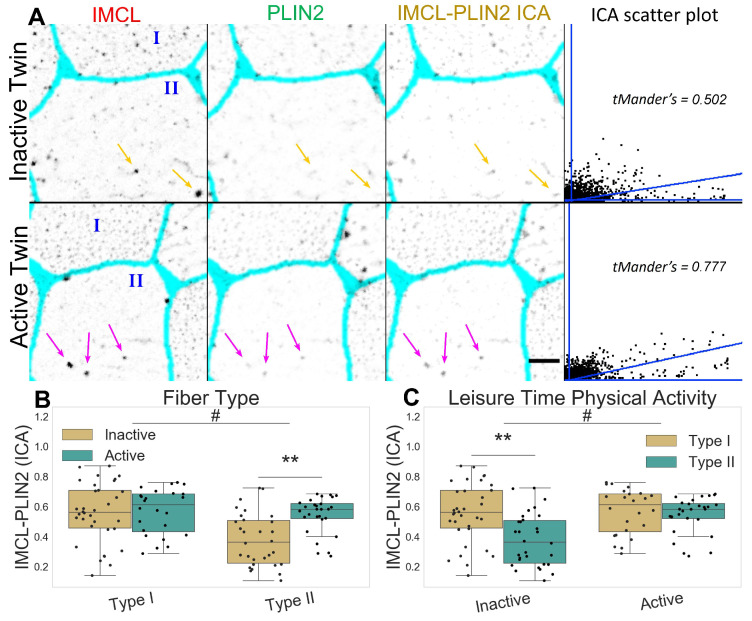
IMCL-PLIN2 intensity correlation analysis in twin pairs. (**A**) representative image showing differences between groups. Gray level indicates marker signal, cyan indicates segmented sarcolemma. Note active twin type II fiber with high intensity IMCL significantly colocalized by PLIN2 (magenta arrows), unlike the inactive twin (orange arrows). Bar = 10 μm. Scatter plot is relative to the arrowed type II fiber; (**B**) fiber type as main effect, with LTPA combined; (**C**) LTPA as main effect, with fiber type combined, main effect differences denoted with **#** (*p*
<0.01); combined group differences denoted with ****** (*p*
<0.001). Dots in (**B**,**C**) represent individual cells.

**Figure 4 ijms-24-04282-f004:**
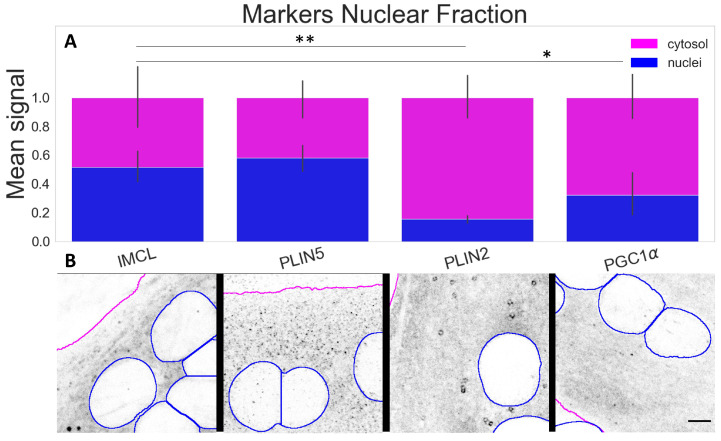
Marker signal comparison between compartments; (**A**) proportion of nuclear signal (blue bars) in relation to cytosolic signal (full bars). Data normalized to the cytosolic reference and measured from the control group (Normal BCAA|Rest). Differences between nuclear fractions of IMCL versus remaining markers ***** (*p* <0.05) and ****** (*p*
<0.001). Whiskers signify standard deviation; (**B**) representative images of respective markers. Gray is signal, blue are limits of segmented nuclei, magenta are limits of segmented myotubes. Bar = 5 μm.

**Figure 5 ijms-24-04282-f005:**
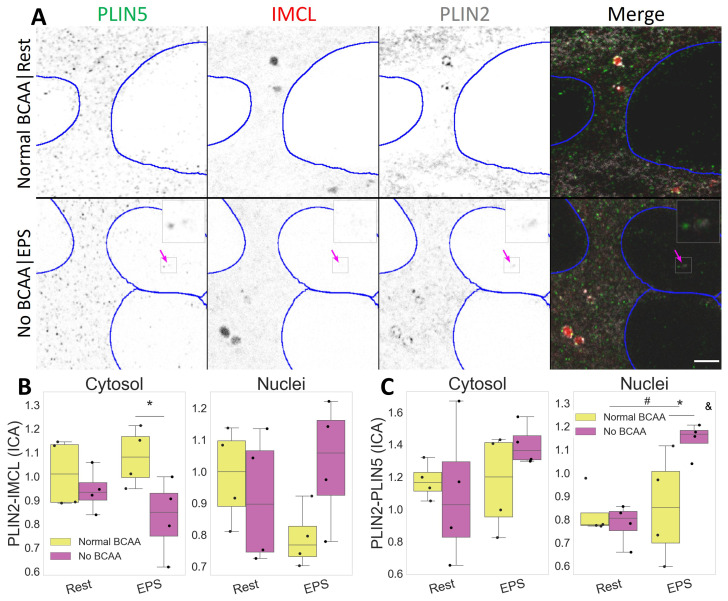
Compartmental PLIN2 association with IMCL and PLIN5 after EPS and BCAA deprivation. (**A**) representative image. Note a more diffused PLIN2 pattern after BCAA deprivation and PLIN2-PLIN5 association in nuclei after EPS (pink arrow). Gray is signal, blue are limits of segmented nuclei. Bar = 3 μm; (**B**) colocalization via intensity correlation analysis (ICA) between PLIN2 and IMCL; (**C**) colocalization via intensity correlation analysis (ICA) between PLIN2 and PLIN5. Main effect differences denoted with **#** (*p*
<0.05). Combined group differences denoted with ***** (*p*
<0.05); interacting effect between fiber type and LTPA denoted with **&** (*p*
<0.05). Dots in (**B**,**C**) represent averaged coverslip values; data normalized to the control group reference (Normal BCAA|Rest).

**Figure 6 ijms-24-04282-f006:**
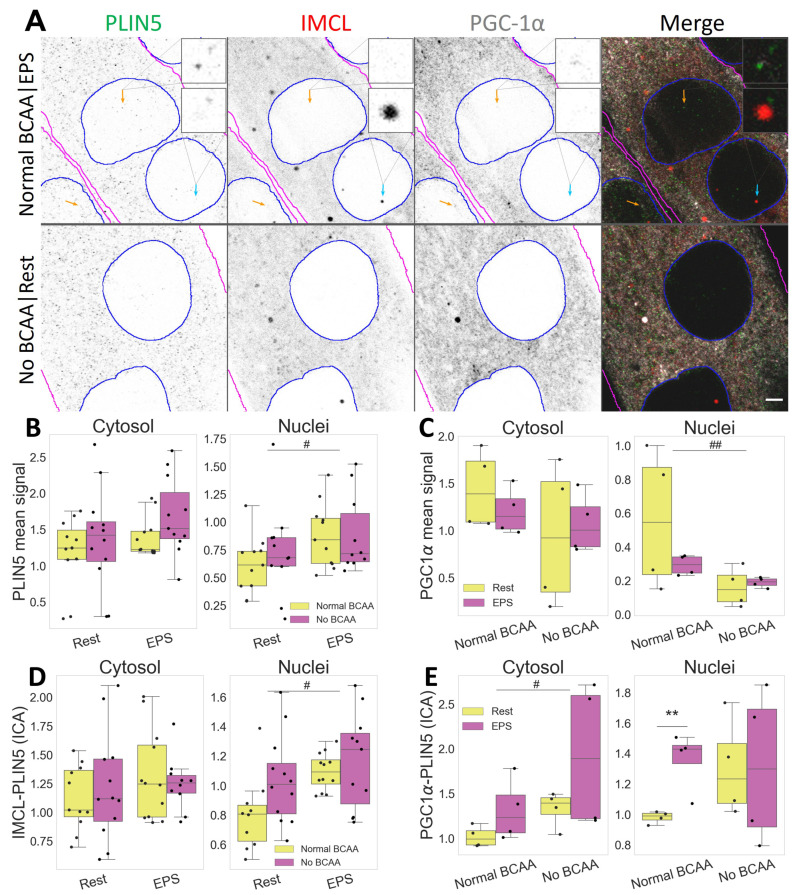
Compartmental association and distribution of PLIN5, IMCL and PGC-1α after EPS and BCAA deprivation. (**A**) representative image. Note more abundant PLIN5 in nuclei after EPS, with stronger association with PGC-1α (orange arrows) and IMCL (cyan arrow). Gray is signal, blue are limits of segmented nuclei, magenta are limits of segmented myotubes. Bar = 3 μm; (**B**) PLIN5 signal intensity in different compartments; (**C**) PGC-1α signal intensity in different compartments; (**D**) ICA between IMCL and PLIN5 in different compartments; (**E**) ICA between PGC-1α and PLIN5 in different compartments. Main effect differences denoted with **#** (*p*
<0.05) and **##** (*p*
<0.01). Combined group differences denoted with ****** (*p*
<0.01). Dots in B–E represent averaged coverslip values; data normalized to the control group reference (Normal BCAA|Rest).

**Figure 7 ijms-24-04282-f007:**
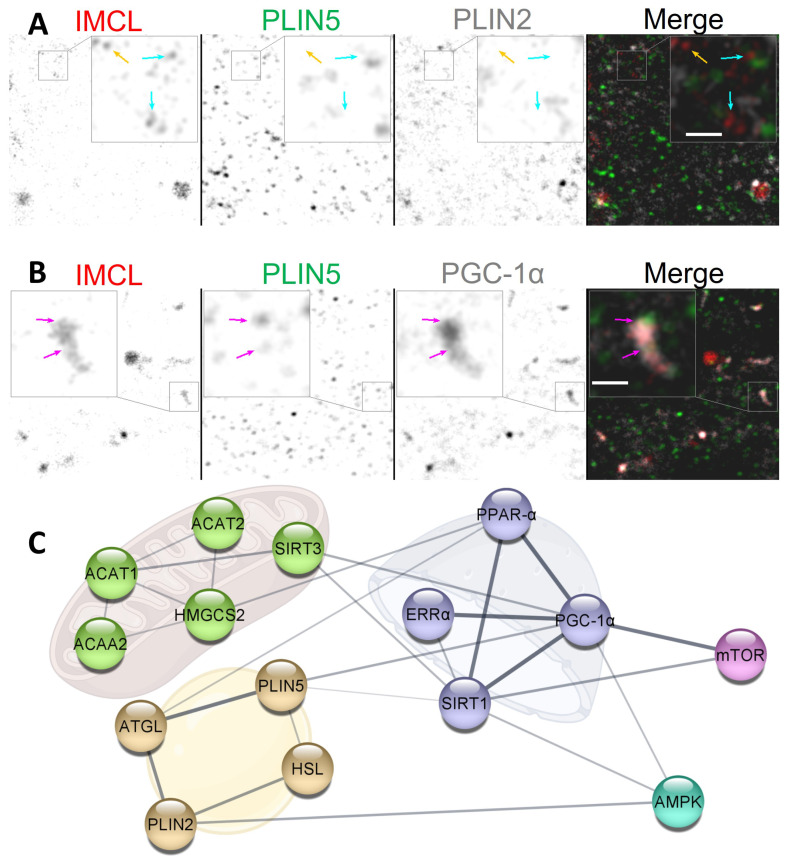
Potential dynamics between IMCL, PLINs and PGC-1α. (**A**) dissociating cytosolic PLIN2. Note that IMCL aggregates with no PLIN2 associated (cyan arrows) and with no PLIN association whatsoever (orange arrow); (**B**) cytosolic association of IMCL-PLIN5-PGC-1α. Note occasional but extensive association between IMCL and PGC-1α, often colocalizing with PLIN5 (magenta arrows). Gray is signal. Bars in (**A**,**B**) =500 nm; (**C**) physical interaction network from *STRING* databases. Golden cluster are LD proteins. Green clusters are mitochondrial enzymes involved in BCAA degradation and cholesterol biosynthesis. Purple clusters are nuclear proteins involved in transcription. Cyan positively responds to exercise. Pink positively responds to BCAA availability. Line thickness indicates the strength of data support. See the end of document for abbreviations.

**Table 1 ijms-24-04282-t001:** Characteristics of twin pairs. Mean ± SEM. ** *p* < 0.001 with *t*-test.

	Inactive	Active
Number of Participants	4	4
LTPA (MET-hours · day^−1^) **	2.9 +/− 1.4	13.8 +/− 1.0
Age (years)	58.0 +/− 2.9	58.0 +/− 2.9
VO_2_ max (mL · min^−1^ · kg^−1^)	30.2 +/− 1.4	32.8 +/− 1.8
Body weight (kg)	71.5 +/− 3.4	69.8 +/− 5.1
BMI (kg · m^−2^)	25.0 +/− 0.6	24.6 +/− 1.1
Body fat (%)	24.1 +/− 2.9	20.2 +/− 3.3
Triglycerides (mmol · L^−1^)	0.9 +/− 0.2	1.0 +/− 0.3
HOMA index	1.9 +/− 0.3	1.5 +/− 0.5

## Data Availability

Data are available upon request. All *ImageJ* and *Python* routines can be found at https://github.com/seiryoku-zenyo/twinC2C12-studies, accessed on 1 August 2022.

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
