# Peer review of "Effects of Long-Term Physical Activity and BCAA Availability on the Subcellular Associations between Intramyocellular Lipids, Perilipins and PGC-1α"

_ijms, 2023, doi:10.3390/ijms24054282_

Round 1
Reviewer 1 Report (Previous Reviewer 2)
The Authors have addressed some of the original comments they did not address during the prior revision round. I would like to thank the Reviewers for their effort.
Reviewer 2 Report (Previous Reviewer 1)
no further comments
This manuscript is a resubmission of an earlier submission. The following is a list of the peer review reports and author responses from that submission.
Round 1
Reviewer 1 Report
The present principal study investigates the influence of physical activity and BCAA availability on components oft he skeletal muscle lipid metabolism. Especially members oft he perilipin family are in the focus oft he analyses. The effects of long-term physical activity on IMCL and perilipins (PLINs) amongst genetically similar individuals is analyzed in skeletal muscle fibres. Moreover the IMCL, PLIN2, PLIN5 and PGC-1α subcellular responses to exercise and BCAA availability is investigated and it is hypothesize that skeletal muscle PLIN5 and PGC-1α signals associate in response to EPS, especially within the nuclei. The study use different methodical approaches which are well selected and state of the art. Especially the confocal laser based analyses are very interesting. Altogether the study over some interesting inside in interplay between functional BCAA metabolism, physical activity and an efficient lipid metabolism. The study has some limitations which are given in the specific comments.
Specific comments:
1. It is a small number of participantes in the twin study therefore the results and interpretation are limited.
2. It is an interesting twin comparison, but it gives the question how long it need up to differences in BCAA/IMCL regulation. Moreover it gives the question if epigenetic mechanisms are involved.
3. It would be helpful to provide also data for the fibre type distribution in inactive and active twins.
4. Is there a difference in IMCL subcellular distribution in the muscle fibres from inactive and active twins?
5. The twin analyses and the in vitro experiments can't be directly compared. In vitro short time effects are investigates which gives an inside how subcellular molecular interaction is changed by EPS and BCAA deprivation. The twin analyses are more focused to long time adaptation, although the time frame for alteration isn’t clear.
6. For future studies it would be interesting to see if short time activity and BCAA deprivation is different in inactive and active twins.
7. It isn't clear from the analyses if it is the effect of life-long physical activity. Therefore the interpretation should be more careful.
Reviewer 2 Report
Review assignment for ijms-1920464-peer-review-v1
In this manuscript, Fachada and colleagues report on intramyocellular lipids (IMCLs), PLIN2 and PLIN5 localization in muscle biopsy sections from twins with opposite lifelong activity grade (active vs sedentary) and in myotubes obtained from C2C12 myoblasts through immunostaining. The comparisons between twins with different grades of long-term physical activity is obviously compelling. However, I think the study should address the following points to expand significance and effectively support the claims:
Major points:
- What is the rationale for BCAA deprivation? This needs to be better explained and introduced. Importantly, what happened to EPS-treated myotubes with BCAA (over)supplementation? What are the levels of expression (mRNA, protein) for the known BCAA transporters in patient biopsies and EPS-treated myotubes?
- the difference between IMCL/PLIN2?PLIN5 staining in Type1 versus Type2 myofibers in inactive vs active twins' biopsies is interesting. How about differences in sub-types of Type2 myofibers (Type 2A, 2X, 2B)?
- can the Authors verify veracity of the IMCL staining through confocal high-magnification images of one representative group of lipid droplets with other markers than PLIN2 and PLIN5? they are using the Bodipy LD540 dye which lends itself pretty well to high-mag confocal imaging...
- were the overall levels of IMCL (triacylglycerols? diacylglycerols?), PLIN2 and PLIN5 different between twin groups? this could be assessed through targeted assays and WBs...
- the nuclear/cytoplasmic fraction analysis performed with imaging is troublesome. the "nuclear" signal could be cytoplasmic signal right underneath or above the nucleus, etc... confocal 3D images and/or side-view images of the representative nuclear/cytoplasmic compartments images are required. This is important because the Authors need to ensure that the IMCL staining they are picking up in the nucleus is real and not an artifact. The point of nuclear/cytoplasmic distribution differences for PLIN2, PLIN5 and PGC1alpha should anyway be compounded with protein fraction WB analyses, particularly from the C2C12 myotubes where material is not limiting.
- a proof of principle to replicate the C2C12-myotube-derived data in primary satellite cells should be considered and reported. C2C12 myoblasts are too far removed from the actual muscle biology to provide a reliable in vitro model as the ONLY in vitro myotube system to expand investigation inspired by human muscle biopsies. Either primary satellite cell-derived myotubes or primary myofibers should be considered, probably the easiest would be from mice.
Minor points:
- The title should be more reflective of the technical limitations of the study and closer to the actual findings. Something like “immunofluorescence in biopsies of activity-divergent twin pairs and in vitro C2C12 myotubes shows different subcellular localization of IMCL, PLIN2, PLIN5 according to activity and BCAA removal” or something along those lines should be considered. As such, the paper is not geared to support any type of mechanistic assessment particularly regarding perilipins, as suggested in the current title.
- the Abstract is very hard to follow and frankly intercalating results from C2C12 myoblasts with patient biopsies results is inappropriate. please consider reformatting the abstract with a clearer, possibly basic-to-translational thought path.
- number of individuals, sex distribution, demographic/physical characteristics, type of biopsy/muscle type, etc... of participants need to be introduced in the Results section before the first results are described!
- there seems to be a confusion regarding nomenclature that needs to be addressed to avoid confusion: if we're talking about the muscle cells within the actual muscle, these should be referred to as "myocytes" or probably a little more appriopately as "myofibers"; if we're talking about C2C12-derived myotubes, those - and only those - should be referred to as "myotubes"
- related to the prior point, in the results section: please divide and clearly delineate which experiments/results are from human muscle biopsies and which results/experiments are from C2C12-derived myotubes!
Round 2
Reviewer 1 Report
No further comments
Reviewer 2 Report
Unfortunately, the Authors did not address satisfactorily the points raised upon first review. Molecular and cellular validations of the staining results are still required to support the claim and frankly significance of the overall work.